# Outcome Expectations and Older Adults with Knee Osteoarthritis: Their Exercise Outcome Expectations in Relation to Perceived Health, Self-Efficacy, and Fear of Falling

**DOI:** 10.3390/healthcare11010057

**Published:** 2022-12-25

**Authors:** Su-Feng Chu, Hsiu-Hung Wang

**Affiliations:** 1Department of Nursing, Meiho University, Pingtung 912009, Taiwan; 2College of Nursing, Kaohsiung Medical University, Kaohsiung 80708, Taiwan

**Keywords:** exercise outcome expectation, knee osteoarthritis, older adults

## Abstract

Outcome expectations are a determinant of exercise engagement and adherence. However, the factors that influence outcome expectations for exercise remain poorly understood for people with knee osteoarthritis. In this paper, a cross-sectional study was conducted by recruiting 211 older adults from three clinics in Southern Taiwan. This study explored older adults with knee osteoarthritis exercise outcome expectations and perceived health, self- efficacy, and fear of falling. The older adults completed the Outcome Expectations for Exercise Scale (OEES), the pain and physical function subscales of Western Ontario and McMaster Universities Osteoarthritis Index (WOMAC), the Perceived Health Status Scale, the Self-Efficacy for Exercise scale (SEE- C), the Activities-Specific Balance Confidence Scale (ABC), the Geriatric Depression Scale (GDS). Multiple logistic regression models were used to determine associations between outcome expectations for exercise and physical and psychosocial outcomes in the knee OA population. Among the participants of the cross-sectional study, the mean age was 72.04 (SD = 5.53) years, and 71.6% were female. Higher outcome expectations for exercise were associated with higher physical function (OR = 0.98; 95% CI [0.96–1.99]; *p* = 0.007), better perceived health (OR = 1.30; 95% CI [1.12–1.51]; *p* < 0.001), greater self-efficacy (OR =1.03; 95% CI [1.01–1.04]; *p* = 0.006), and less fear of falling (OR = 3.33; 95% CI [1.21–9.19]; *p* = 0.020). Thus, the results indicated that outcome expectations for exercise among the participants were significantly associated with physical function, perceived health, self-efficacy, and fear of falling. These findings suggest the importance of personal factors in the design of interventions to promote exercise behavior changes among elderly patients with Knee Osteoarthritis.

## 1. Introduction

Knee osteoarthritis (OA) is one of the most common degenerative joint illnesses in older adults. It is commonly associated with joint pain, difficulty in daily activities, impaired physical function, increased risk of falling, and depression [1,2,3]. Exercise is the most recommended non-drug treatment to reduce the continued deterioration of joint symptoms [4]. In addition, exercise can improve the patient’s quality of life and reduce the need for drugs and surgery [5]. While the benefits of exercise have been clearly demonstrated, participation in exercise remains a challenge for patients with knee OA [6]. People with knee OA seem to exercise less frequently than the general population [7], and exercise adherence is a common problem among these people [8]. A recent meta-analysis, in which the participants’ mean age was 64 years, reported that only 13% of patients with knee OA met physical activity guidelines [6]. Therefore, there is an urgent need to identify the factors that influence exercise behavior in older adults with knee OA.

Numerous personal factors are associated with exercise in knee OA, including age, gender, education level, the perceived severity of symptoms, physical and mental health, self-efficacy, outcome expectations and motivation [8,9,10]. Among these factors, outcome expectation and self-efficacy are essential constructs of Bandura’s social cognitive theory, which forms exercise intentions, influences the motivation to participate and adhere to exercise, and can significantly predict personal exercise behavior [11,12,13].

Most studies have focused on the topics of self-efficacy and exercise behavior [11,13,14,15,16]; however, several studies have also reported that outcome expectations are better predictors of exercise behavior than self-efficacy [17,18]. In clinical settings, a growing body of research supports the claim that outcome expectations affect exercise behavior in diverse populations [12,19,20]. Based on recent findings, the use of outcome expectations is strongly recommended as a significant predictor of exercise behavior in older adults [21,22].

Outcome expectations refer to when individuals decide to take action based on personal belief and their expectation that the pursued activity will produce desired outcomes [13]. For example, when individuals have a higher degree of perceived physical and psychological benefits (high outcome expectations), there is a positive influence on exercise behavior [23,24]. Therefore, those with higher outcome expectations for exercise are more motivated to exercise than those who perceive fewer benefits (lower outcome expectations) [20,25,26]. In the knee OA population, this growing body of literature has proven that the outcome expectations of patients with knee OA are a crucial predictor of exercise behavior [9,24,27]. However, despite substantial evidence supporting the predictive role of outcome expectations in exercise behavior, the precise influencing factors that influence outcome expectations for exercise remain poorly understood. 

Studies on the relationships between outcome expectations for exercise and demographic variables such as age, gender, and education level have yielded inconsistent results [23,28,29,30,31]. The factors that have been demonstrated to influence outcome expectations for exercise include the patient’s individual physical and psychosocial health, such as their personal perception of health status [20,32,33], self-efficacy [12,34], pain [33,34], fear of falling [33,35,36] and depression [12,20,26,37]. However, in general, the samples used largely targeted older adults and community settings. Moreover, most of these studies have explored or tested the construction of a theory or model, and did not directly assess influencing factors related to outcome expectations for exercise. One qualitative study focused on knee pain in older adults, determining that higher exercise outcome expectation were also associated with well-being, mood, self-efficacy, and personal past experiences [34]. However, these studies did not explore exercise outcome expectations in older adults with knee OA. 

A recent clinical study reported that higher outcome expectations for exercise in people with knee OA were associated with self-efficacy and depression, but not pain, physical function, or education levels [24]. These findings are inconsistent with most previous studies on pain [20,27,34], and physical function [20,38]. In addition, the study did not discuss influencing factors related to outcome expectations for exercise, such as fear of falling, nor did it examine the relationship between patients’ perceived health statuses and their outcome expectations for exercise. Furthermore, the study recruited participants between the ages of 40 and 70.3, with a mean age of 59.8, an age group skewed toward adults and younger seniors who may have different outcome expectations for exercise. Moreover, their statistical analysis used a univariate logistic regression model, which cannot control for confounding factors. 

This study sought to explore the relationship between outcome expectations for exercise and psychosocial factors in older adults with knee osteoarthritis. We hypothesized that better physical health (i.e., less pain, higher physical function, and better perceived health status) and better psychosocial health (i.e., higher self-efficacy, less fear of falling, depression) would be positively associated with higher outcome expectations for exercise. The results of the study would be helpful in understanding the individual factors that affect the outcome expectations for exercise in older adults with knee OA. As a result, interventions could be effectively created and implemented to promote exercise behavior changes among elderly patients with knee OA, which would ease their joint discomfort and reduce the necessity of making medical interventions. 

## 2. Materials and Methods

### 2.1. Study Design and Participants

This study adopted a cross-sectional design and purposive sampling. Participants were older adult patients diagnosed with knee OA at orthopedic clinics in three hospitals in Southern Taiwan. The inclusion criteria of this study were as follows: (a) diagnosed with knee OA by a physician; (b) aged 65 years or older; (c) conscious and able to verbally communicate in Mandarin or Taiwanese; and (d) agreed to participate in the study after it was explained. Patients were excluded if they (a) had physical disabilities resulting from diseases other than knee OA; (b) had accepted knee OA reconstruction surgery; (c) were diagnosed with cancer; or (d) had any cognitive or mental illness.

Study interviewers were trained registered nurses, and these nurses conducted one-on-one interviews. Because participants were older people, study interviewers slowed their speaking speed during interviews, and provided participants with sufficient time to think about and answer the questions. Study interviewers read and explained the questions and carefully guided participants in reading the questions on paper to increase their understanding of the content of the questions. The interview process took approximately 30 min. Previous research data were used as references for the parameters [24]. G*Power v3.1.7 was applied to determine a sufficient sample size by using an alpha of 0.05, a power of 0.80, a small effect size (odds ratio (OR) = 1.05), and a two-tailed test. Based on the aforementioned assumptions, the minimum sample size of 168 was determined. In total, 211 older adults were recruited and completed the study questionnaire. 

### 2.2. Measures

#### 2.2.1. Outcome Expectations

Outcome expectations for exercise (OEE) is a measurement with in the Outcome Expectations for Exercise Scale (OEES). Lee et al. translated the original scale into a Chinese version to measure exercise outcome expectations in patients with OA. The original OEE scale was developed by Resnick et al. in 2001 for older populations. This scale was designed to evaluate the level of benefits older people receive as a consequence of their beliefs concerning exercise behaviors, and comprises nine questions. A 5-point Likert scale is used, with scores ranging from 1 (strongly disagree) to 5 points (strongly agree). The total score ranges from 9 to 45 points, with higher scores indicating more positive exercise outcome expectations. Cronbach’s alpha coefficient of the OEES is 0.80 [39]. The OEES’s dependent variable is not a normal distribution and is skewed toward a high range, with a median of 36; therefore, based on the point at which the group we studied could be divided in two, we applied a binary variable for the OEES. An OEES score of <36 was a low OEES, and an OEES score of ≥36 was a high OEES.

#### 2.2.2. Pain and Physical Function

Pain and physical function were measured using two WOMAC subscales (Western Ontario and McMaster Universities Osteoarthritis Index). The scale was developed by Bellamy et al. in 1998. This constitutes a self-evaluation scale, and is widely used to assess OA symptom severity. This pain scale evaluates activity pain or rest in the last 48 h and employs five items (with a total score of 0–50 points). Higher scores indicate more severe pain. The WOMAC physical function subscale, which comprises 17 items, assesses difficulty in performing physical activities (for a total score of 0–170 points). Higher scores indicate poorer physical functions. Cronbach’s alpha is 0.97. The construct validity was tested by correlating WOMAC items with a visual analog scale of pain and handicaps that present significant correlations (*p* < 0.01) [40].

#### 2.2.3. Perceived Health Status

Perceived health status refers to the self-perceived health status of an individual, i.e., the individual’s self-evaluation of their own health conditions. To measure health status, this study used the Chinese version of the Perceived Health Status Scale, which was revised by Huang and Chiu from the General Health Perception subscale of the SF-36 Health Status Scale developed by Ware et al. in 1994. The scale provides a one-year-long self-evaluation of health conditions. In total, there are 3 questions rated from 1 to 5 points, with 1 point indicating extremely good health conditions and 5 points indicating extremely poor health conditions. The Huang and Chiu’s scoring system ranges from 3 to 15 points and is reversed, with higher scores indicating better self-evaluated health conditions. The Cronbach’s alpha value is 0.84 [41].

#### 2.2.4. Self-Efficacy

Self-efficacy refers to subjective judgments of whether an individual can successfully carry out a particular behavior. This study used Lee et al.’s Chinese translation of the Self-Efficacy for Exercise scale (SEE-C), which was originally developed by Resnick and Jenkin in 2000 to evaluate exercise self-efficacy.

The scale contains nine questions, and is used to measure older adults’ levels of confidence in exercising under various circumstances that are unfavorable to exercise. The score ranges from 0 (completely unconfident) to 10 points (completely confident). The total score ranges from 0 to 90 points, with higher scores indicating higher self-efficacy. Cronbach’s alpha of this scale is 0.75. In our study, for criterion-related validity, results indicating exercise self-efficacy were positively correlated with physical activity level (*r* = 0.46; *p* < 0.0001) [15].

#### 2.2.5. Fear of Falling

To accurately measure the fear of falling, the Activities-specific Balance Confidence Scale (ABC) was developed by Powell and Myers in 1995. This study used Hus and Miller’s Chinese translation of the ABC to evaluate older people’s confidence about their balance in executing indoor and outdoor activities, including walking through parking lots, traveling up and down inclines, mopping the floor, walking up and down stairs, and picking up slippers from the floor. The scale comprises 16 questions, with total scores ranging from 0 (completely unconfident) to 100 points (completely confident). Scores lower than 50 points indicate lower confidence in balance functions. The test–retest reliability of this scale is 0.88 (95% CI, 0.78–0.94) [42]. In this study, scores lower than 50 points indicated lower confidence in balance functions, and scores higher than 50 points indicated higher confidence. In addition, higher scores indicated less fear of falling.

#### 2.2.6. Depression

The Geriatric Depression Scale (GDS) was initially developed by Yesavage et al. in 1983. In 1986, a short questionnaire comprising 15 questions related to depression symptoms was also developed, and it has been extensively applied to older populations. This study used the translated Chinese version of the GDS-Short Form (GDS-SF), which contains 15 questions. Total scores of 0–4 points indicate no depression, 5–8 points indicate mild depression, 9–11 points indicate medium depression, and 12–15 points indicate major depression. Moreover, higher scores indicate more severe depression. In this study, the scale was used on Chinese people, and its sensitivity and specificity were 96.3% and 87.5%, respectively [43].

### 2.3. Ethical Considerations

This study was conducted in accordance with the Declaration of Helsinki and approved by the Institutional Review Board of Kaohsiung Medical University (reference number: KMUHIRB-(I)-20180289; date of approval: 24 October 2018). All participants were informed of voluntary participation and guaranteed confidentiality. In addition, we were provided informed consent, and agreements were completed for the questionnaire survey.

### 2.4. Data Analysis

Data were analyzed using SPSS statistical software version 23.0. Means, standard deviations, and frequencies are listed for descriptive statistics. The OEES, as a dependent variable, did not have a normal distribution and was skewed toward the higher range, with a median of 36; consequently, participants were divided into two groups. As noted above, an OEES score of <36 was categorized as a low OEES, and an OEES score of ≥36 was deemed a high OEES. Lower and higher outcome expectation groups were compared using an independent *t*-test (degree of freedom = 1) for continuous variables (i.e., pain, function, perceived health status, self-efficacy, depression) and chi-square tests for categorical variables (i.e., fear of falling). Multiple logistic regression was employed to explore the association between participant characteristics and high outcome expectations. A two-sided *p*-value of <0.05 was considered significant.

## 3. Results

### 3.1. Participants’ Demographic Information and Research Variable Date

Table 1 lists the participants’ demographic information and research variable data. The mean age of the 211 participants was 72.04 (SD = 5.53) years; most of them were female (71.6%), and 76.3% had a junior high school or lower level of education. The OEES score was 33.35 (SD = 8.56). The WOMAC pain and function subscale scores were 15.03 (SD = 14.42) and 44.49 (SD = 38.71), respectively. The perceived health status score was 9.33 (SD = 3.16), the SEE-C score was 36.58 (SD = 26.84), the ABC scale score was 68.40 (SD = 23.87), and the GDS-SF. score was 6.05 (SD = 1.68).

### 3.2. Results of Multiple Logistic Regression Analysis to Predict High OEE

Table 2 demonstrates that higher outcome expectations were significantly associated with female, high educational levels, regular exercise, high physical function, perceived good health status, high self-efficacy and less fear of falling. The OR [95% confidence interval (95% CI)] were 2.95 [1.18, 7.40], 5.21 [1.39, 19.45], 5.38 [1.79, 16.16], 0.98 [0.97, 0.99], 2.27 [1.44, 3.59], 1.03 [1.01, 1.04] and 3.69 [1.33, 10.24], respectively. Results for factors related to exercise outcome expectation are also presented in Figure 1.

### 3.3. Difference between Lower and Higher Outcome Expectation Groups

Table 3 presents the differences in the means of variables between the lower and higher outcome expectation groups. Significant differences were identified in the mean WOMAC pain scores, the function scores, the perceived health status scores, the SEE-C scores, and the ABC scores between lower outcome expectations and higher outcome expectations. All *p*-values were <0.001, and the GDS-SF score was *p* < 0.044.

## 4. Discussion

As hypothesized, our findings revealed that higher outcome expectations were significantly associated with gender, regular exercise, higher educational levels, higher physical function, better perceived health status, higher self-efficacy, and less fear of falling. However, age, chronic illness, fall experience, pain, and depression were not found to be significantly associated with exercise outcome expectations. 

The finding that gender is a significant predictor of outcome expectations generally supports the results of prior research [23,28]. However, the precise contribution of gender remains somewhat unclear since one of the previous investigations found—as we did—that females possess higher exercise outcome expectations, whereas another study found that males possess higher exercise outcome expectations [32]. Furthermore, yet another study on individuals with knee OA showed no significant association between gender and exercise outcome expectations [24]. 

Our study suggests that a higher educational level is significantly associated with higher outcome expectations. The results are consistent with previous studies on other populations [44,45]. However, another recent study on knee OA population rejected an association between outcome expectations for exercise and level of education [24]. In support of our finding, a study conducted by Resnick et al. demonstrated that individuals afflicted with osteoporosis who are more educated may have a better understanding of the benefits of exercise [45]. In other words, those with higher levels of education could develop higher outcome expectations for exercise. Therefore, it appears elderly knee OA patients with lower education level have a more pressing need to understand the benefits of exercise to enhance their outcome expectations. 

Patients who reported that they have a habit of regularly exercising have a significantly higher outcome expectations than those who did not, which is consistent with findings in other populations [28,32,46]. Many studies have reported that regular exercise is effective in reducing pain and improving physical function in arthritis patients [5,47,48] and that those who engaged in it had better perceived health and functional statuses [46]. Individuals with knee OA who opt to exercise can expect reductions in knee pain and all-cause disability as the years progress, and they also have higher outcome expectations for exercise. This also confirms the social cognitive theory that outcome expectations are influenced by past experience [49]. 

Based on the above discussion, our study shows that personal characteristics, such as gender, education level, and regular exercise, influence outcome expectations for exercise. This finding drastically departs from the results of another recent study, which found that it may not be possible for gender and educational level to influence outcome expectations for exercise in knee OA populations [24]. There are limited studies in the literature involving the association between personal characteristics and outcome expectations for exercise, especially in knee OA populations. To shed more light on this issue, further research is needed on knee OA, especially among older adults. 

Our results show that OA-specific variables of pain are not associated with higher outcome expectations for exercise, similar to what was reported in another recent study on patients with knee OA [24]. These participants, however, were recruited in clinical settings, and may have received encouragement to exercise in order to relieve pain. In addition, the clinics may have taught these participants about pain self-regulation [50], as well as how to manage symptoms and have better pain coping skills [51]. Consequently, we suggest that pain may not influence exercise outcome expectations in knee OA populations who actively seek professional assistance. In contrast, numerous community resident studies have had different results, for example, in one arthritis population less knee pain caused participants to have higher outcome expectations for exercise [27,34]. Based on these studies on community resident, it can be seen that the presence of pain is related to outcome expectations for exercise. Therefore, further studies comping knee OA populations in clinical and community samples are needed. 

Good physical function was found to be significantly related to higher outcome expectations. Similarly, a recent study found that dialysis patients with better physical function possessed better exercise outcome expectations [38]. In contrast, a study on patients with knee OA found no relationship between physical function and outcome expectations for exercise [24]. These disparate results may be attributed to the fact that the majority of the study participants skewed toward a younger age, which may not be representative of the older adult population. Physical function and pain constitute major, OA-specific variables in assessing symptom severity [1,52]. We found that the OA-specific variable of physical function is an indicator of exercise outcome expectations the in knee OA population. This indicates that the physical function variable is not only associated with OA symptom severity but also outcome expectations for exercise in older adult with knee OA. Knee OA often results in functional limitations. Improving physical function to enhance exercise outcome expectations and motivate exercise engagement in this population is highly recommended. 

Our research determined that perceived health status is a significant indicator of exercise outcome expectations among people with knee OA, which is consistent with the findings of other studies [20,32,53]. However, few studies have examined the link between perceived health status and outcome expectations for exercise using homogeneous samples. A recent study found that healthier and physically stronger patients possessed better exercise outcome expectations [38]. According to an arthritis patient study, 71.8% of participants self-reported average or poor health levels [54]. Therefore, attention must be paid to the impact of perceived health status on outcome expectations for exercise especially among older adults with knee OA. 

We also found a significant association between participants with higher self-efficacy and higher exercise outcome expectations. According to social cognitive theory, self-efficacy and outcome expectations are important constructs that influence people to adopt and maintain exercise behavior. Self-efficacy operates in conjunction with outcome expectations to influence exercise behavior in older adults [55]. In support of this theory, previous clinical studies showed that knee OA populations with higher outcome expectations are significantly more likely to have more confidence in their ability (higher self-efficacy) to exercise [21,46]. Our findings are consistent with this theory and confirms that self-efficacy is related to exercise outcome expectations among people with OA. Therefore, we suggest that exercise interventions must consider both self-efficacy and outcome expectations to enhance exercise participation and persistence in patients with knee OA. That said, another study found that stereotypical beliefs about arthritis(e.g., that arthritis is a natural part of life, and once you are diagnosed with arthritis, it can only become worse) [56] and aging (e.g., people should expect to live in pain and not be able to walk as well with age) may produce negative outcome beliefs, which reduce self-efficacy [57] and negatively affect outcome expectations. In the future, additional studies that focus on these aspects are warranted.

Falling and fear of falling are often present in older adults with knee OA. We found that maladaptive psychosocial factors such as fear of falling are a determinant of exercise outcome expectations among people with knee OA. As mentioned earlier, although falls do not affect outcome expectations for exercise, fear of falling adversely impact patients’ outcome expectation and their motivation to exercise [58]. A previous qualitative study indicated that an increased fear of falling may cause negative outcome expectations for exercise [36,59]. Recent clinical studies on osteoporosis and post-hip-fracture populations also found that fear of falling is negatively associated with higher exercise outcome expectations [45,53]. With their similar results, these qualitative and clinical studies support our findings. Studies have shown that patients with severe knee OA lose self-efficacy and self-confidence, avoidance of activities, and lower outcome expectations because of their fear of falling [58,60]. Therefore, the relationship between exercise outcome expectations and fear of falling may vary depending on the severity of knee OA. Further research is needed to corroborate this theory. 

We did not find that depression was significantly associated with exercise outcome expectations, although other previous studies suggest that depressed individuals are more inclined to hold a pessimistic attitude about exercise outcomes [24,26,32,37]. These studies showed that patients with depressive moods exhibit a motivational deficit and have lower outcome expectations for exercise. That said, a statistically significant relationship was not found between depression and outcome expectations in our study; in part, this may be due to a lack of variance in depression, with the majority of participants having no depression or mild depression. However, it is still unclear if this association would remain in a more depressed population, especially in a knee OA population. Therefore, further studies are necessary to understand the relationship between different levels of depression and exercise outcome expectations. 

## 5. Limitations

The present study contains certain limitations. First, our study used information that was self-reported by older participants, which might be subject to memory or attention bias. They may have overestimated or underestimated exercise outcome expectations. Second, the current literature on exercise outcome expectations is limited. More research is required before we can draw definitive conclusions regarding the influence of factors on outcome expectations for exercise. Third, most participants are retired, giving them greater flexibility and time to exercise which benefits them. In addition, we recruited participants seeking knee OA treatment from clinics who were more motivated and had higher outcome expectations for exercise than community-dwelling elderly individuals. Fourth, our study used a cross-sectional research design, and causal inferences and trends from problems or phenomena could not be elucidated from the findings. However, overall, despite these constraints, the study results are relevant to clinical settings and suggest productive directions for further research.

## 6. Conclusions

Our study findings indicate that older adult’s higher exercise outcome expectations are significantly associated with higher physical function, better health status, higher self-efficacy, and less fear of falling. Higher exercise outcome expectations were also related to personal characteristics such as gender, regular exercise, and higher educational levels. Surprisingly, age, chronic illness, fall experience, pain, and depression were not significantly associated with exercise outcome expectations. Regardless, these findings suggest that personal physical and psychosocial health factors of older adults with knee OA influence their outcomes expectations for exercise.

## 7. Implications

By discovering new and significant relationships, this study clarifies and advances the current understanding of exercise beliefs among older adults with knee OA. These relationships may help distinguish older patients with knee OA at a high risk of lack of exercise or a non-adherence to exercise. In addition, we found that males and less-educated older adults have lower outcome expectations for exercise, and they require effective strategic targeting to improve their exercise outcome expectations. Depression and pain level may have different impacts on outcome expectations in knee OA populations, and further study is needed. Future research must focus on interventions involving the application of these related factors. This is necessary to determine whether improving exercise outcome expectations is effective in enhancing the exercise behavior of older adults with knee OA.

## Figures and Tables

**Figure 1 healthcare-11-00057-f001:**
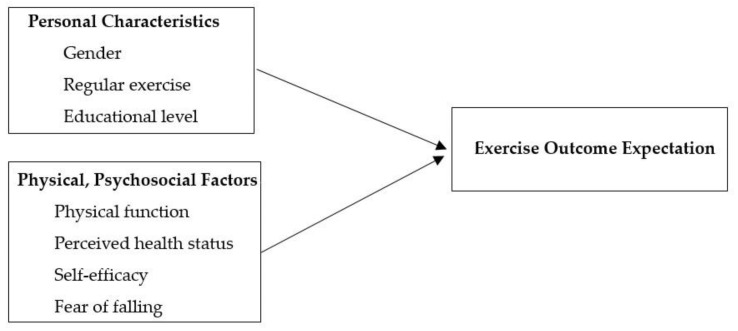
Result of Factors Related to Exercise Outcome Expectations.

**Table 1 healthcare-11-00057-t001:** Participants’ demographic information and research variable Data.

Demographic Variable	Total Sample (n = 211)
Gender	
Male	60 (28.4%)
Female	151 (71.6%)
Age	72.04 ± 5.53
Education status	
Uneducated	45 (21.3%)
≤Junior high school	116 (55%)
≥High school or over	50 (23.7%)
Chronic illness	
With	196 (92.9%)
Without	15 (7.1%)
Fall experience (past year)	
Yes	53 (25.1%)
No	158 (74.9%)
Exercise habits	
Never	39 (18.5%)
Sometimes	56 (26.5%)
Regular	116 (55.0%)
WOMAC pain subscale (range 0–50)	15.03 ± 14.42
WOMAC ^a^ function subscale (range 0–170)	44.49 ± 38.71
Perceived health status (range 3–15)	9.33 ± 3.16
SEE-C ^b^ scale (range 0–90)	36.58 ± 26.84
ABC ^c^ scale (range 0–100)	68.40 ± 23.87
ABC < 50	49 (23.2%)
ABC ≥ 50	162 (76.8%)
GDS-SF ^d^ (range 0–15)	6.05 ± 1.68

a WOMAC = Western Ontario and McMaster Universities Osteoarthritis Index; higher function scores indicate weaker physical activity function; b SEE-C = Chinese version of the Self-Efficacy for Exercise scale; c ABC = Activities-specific Balance Confidence scale; d GDS-SF = Chinese version of the Geriatric Depression Scale—Short Form. Higher scores indicate more severe depression.

**Table 2 healthcare-11-00057-t002:** Results of multiple logistic regression analysis to predict high OEE.

Independent Variable	OR [95%CI]	*p*-Values
at baseline		
Age	1.02 [0.95, 1.10]	0.595
Gender		0.021 *
Male	Ref	
Female	2.95 [1.18, 7.40]	
Education status		
Uneducated	Ref	
≤Junior high school	2.93 [1.09, 7.93]	0.034 *
≥High school or over	5.21 [1.39, 19.45]	0.014 *
Chronic illness		0.414
With	Ref	
Without	1.95 [0.39, 9.60]	
Fall experience (past year)		0.718
Yes	Ref	
No	1.18 [0.48, 2.92]	
Exercise habits		
Never	Ref	
Sometimes	1.86 [0.57, 6.07]	0.306
Regular	5.38 [1.79, 16.16]	0.003 *
WOMAC pain subscale	1.00 [0.96, 1.04]	0.939
WOMAC function subscale	0.98 [0.97, 1.00]	0.021 *
Perceived health status	2.27 [1.43, 3.59]	0.001 *
SEE-C scale	1.03 [1.01, 1.04]	0.007 *
ABC scale		
ABC < 50	Ref	0.012 *
ABC ≥ 50	3.69 [1.33, 10.24]	
GDS-SF ^d^	1.08 [0.85, 1.35]	0.54

* *p* < 0.05 OR = odds ratio 95% CI = 95% confidence interval. d GDS-SF = Chinese version of the Geriatric Depression Scale—Short Form. Higher scores indicate more severe depression.

**Table 3 healthcare-11-00057-t003:** Difference Between Lower and Higher Outcome Expectation Groups.

Demographic	Low OEE	High OEE	*p*-Values
	group (*n* = 97)	group (*n* = 114)	
WOMAC pain subscale (range 0–50)	19.36 ± 12.64	11.35 ± 14.86	^a^ 0.001 *
WOMAC function subscale (range 0–170)	61.93 ± 40.85	29.65 ± 29.75	^a^ 0.001 *
Perceived health status (range 3–5)	8.07 ± 2.84	10.4 ± 3.03	^a^ 0.001 *
SEE-C scale (range 0–90)	24.67 ± 19.92	46.71 ± 27.85	^a^ 0.001 *
ABC scale (range 0–100)			
ABC < 50	38 (39.2%)	11 (9.6%)	^b^ 0.001 *
ABC ≥ 50	59 (60.8%)	103 (90.4%)	
GDS-SF (range 0–15)	6.30 ± 1.84	5.83 ± 1.49	^a^ 0.044 *

*p* relates to the difference in baseline characteristics between the two outcome groups; * *p* < 0.05. a: independent *t*-test; b: X^2^
_df = 1_ test.

## Data Availability

Additional data are available upon request from the corresponding author.

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
