# Peer review of "Outcome Expectations and Older Adults with Knee Osteoarthritis: Their Exercise Outcome Expectations in Relation to Perceived Health, Self-Efficacy, and Fear of Falling"

_healthcare, 2022, doi:10.3390/healthcare11010057_

Round 1

Reviewer 1 Report

The paper presents the relationship between outcome expectations and other related factors, such as physical function, pain, perceived health, self-efficacy, depression and fear of falling, in a specific population as people with knee osteoarthritis (OA).

It offers new results regarding this issue, revealing new relationships between some of these factors.

The topic is relevant and offers new data on a topic under study. The results could have impact in future approaches of exercise adherence strategies in people with knee OA.

In general, the article is well written, but it would need a spell check and a review of some expressions and sentences, that seem to be too long.

The abstract should include the factors that were studied and the measurement tools used in the study.  

Discussion should be revised, as it includes aspects not mentioned before, like mindfulness. If that is a relevant issue, it should be mentioned in the introduction.

Author Response

Thank you for reviewing and giving us the opportunity to revise this manuscript. We have taken into careful consideration each point raised by the reviewers during revision.

Based on reviewers’ comments, we are submitting a revised manuscript and a table with three columns to address all reviewer comments and suggestions. Changes in the manuscript are presented in the red color.

We thank you for the opportunity to submit this revised manuscript again. We look forward to your comments.

Reviewer 2 Report

Thank you for the opportunity to review this interesting manuscript. It is about the understanding of the relationship between expectations for exercise and physical and psychosocial factors in people with OA. The manuscript is well presented and structured and its methodology is strong. This manuscript require minor revisions and I strongly suggest a revision from an English mother language speaker.

Please, clearly write the objective of the study in the abstract

 “mean age of 72.04 (SD = 5.53)”, please, specify that the measure is in years

“Exercise is the most recommended non-drug treatment to reduce continued deterioration of joint symptoms”, please, add a reference, I suggest to read one of the last published articles on this topic “How Physical Activity Affects Knee Cartilage and a Standard Intervention Procedure for an Exercise Program: A Systematic Review”

“Although most of studies have a focus on the topics of self-efficacy and exercise [10, 12-15],however, several studies reported that outcome expectations were better predictors of exercise behavior than self-efficacy was[16, 17].” Please, clarify this sentence, in the present form is not so clear

“However, these findings were inconsistent with previous most of studies on the influence of pain [20, 27, 34]”. Please, clarify this sentence, in the present form is not so clear

“Moreover, this statistical analysis uses a”, I suggest, instead of this, to use the term “their”, “the study”…

“The results of the study are helpful to understand the individual factors which affect the outcome expectations for exercise in older adults with knee OA.”, I suggest “the results of the study could be helpful…”

Table 1. Please, specify that the number within () are in percentage (%)

The results are well structured but I strongly suggest to include a figure to summarize the findings.

Author Response

(The authors gave the same response as above.)
